# Exposure to Phthalate and Organophosphate Esters via Indoor Dust and PM10 Is a Cause of Concern for the Exposed Saudi Population

**DOI:** 10.3390/ijerph18042125

**Published:** 2021-02-22

**Authors:** Nadeem Ali, Nabil A. Alhakamy, Iqbal M. I. Ismail, Ehtisham Nazar, Ahmed Saleh Summan, Syed Ali Musstjab Akbar Shah Eqani, Govindan Malarvannan

**Affiliations:** 1Centre of Excellence in Environmental Studies, King Abdulaziz University, Jeddah 21589, Saudi Arabia; iismail@kau.edu.sa (I.M.I.I.); asumman@kau.edu.sa (A.S.S.); 2Pharmaceutics Department, Faculty of Pharmacy, King Abdulaziz University, Jeddah 21589, Saudi Arabia; nalhakamy@kau.edu.sa; 3Department of Chemistry, Faculty of Science, King Abdulaziz University, Jeddah 21413, Saudi Arabia; 4Department of Environmental Sciences, University of Gujrat, Punjab 50700, Pakistan; chakrian2010@gmail.com; 5Department of Environmental Sciences, King Abdul Aziz University, Jeddah 21589, Saudi Arabia; 6Public Health and Environment Division, Department of Biosciences, COMSATS Institute of Information Technology, Islamabad 45550, Pakistan; ali_ebl2@yahoo.com; 7Toxicological Centre, University of Antwerp, Universiteitsplein 1, 2610 Wilrijk, Belgium

**Keywords:** organophosphate esters, phthalates, PM10, indoor dust, Saudi Arabia

## Abstract

In this study, we measured the occurrence of organophosphate esters (OPEs) and phthalates in the settled dust (floor and air conditioner filter dust) and in suspended particulate matter (PM10) from different microenvironments (households (*n* = 20), offices (*n* = 10) and hotels (*n* = 10)) of Jeddah, Saudi Arabia. Bis (2-Ethylhexyl) phthalate (DEHP) was the major pollutant (contributing >85% of total chemicals burden) in all types of indoor dust with a concentration up to 3,901,500 ng g^−1^. While dibutyl phthalate (DBP) and DEHP together contributed >70% in PM10 (1900 ng m^−3^), which indicate PM10 as a significant source of exposure for DBP and DEHP in different Saudi indoor settings. Tris (1-chloro-2-propyl) phosphate (TCPP) was the major OPE in PM10 with a concentration of up to 185 ng m^−3^ and the occurrence of OPEs in indoor dust varied in studied indoor settings. The estimated daily intake (EDI) of studied chemicals via dust ingestion and inhalation of PM10 was below the reference dose (RfD) of individual chemicals. However, estimated incremental lifetime cancer risk (ILCR) with moderate risk (1.5 × 10^−5^) for Saudi adults and calculated hazardous index (HI) of >1 for Saudi children from DEHP showed a cause of concern to the local public health.

## 1. Introduction 

Organophosphate esters (OPEs) and phthalates are used in various consumer products to add elasticity and retard the onset of unwanted fire to minimize fire hazards [1,2,3]. These chemicals are added in many consumer products to fulfill the safety regulations, e.g., building materials, including thermal insulation boards, electrical and electronic equipment, furniture foams, children’s toys, fabrics and printed circuit boards [4,5,6,7]. Most of these chemicals are added to the products instead of chemically reacted; thus using these products, chemicals released into the human inhabitants and linked with severe health concerns, e.g., reproductive health and endocrine disruptions [5,8,9,10]. Environmental contamination to these chemicals results in different human exposure scenarios via various exposure sources, e.g., food, water, air, dust, etc. Several studies have reported that unintentional intake of contaminated dust and air through inhalation, ingestion and dermal contact are considered essential exposure routes for many of these chemicals [2,3,5,11].

Recent studies have reported these chemicals’ presence in indoor dust and other environmental samples [1,6,7,12,13,14,15,16]. Studies on the presence of these chemicals in the indoor environments of Saudi Arabia are lacking, and only a couple of studies have reported them in indoor dust samples [7,16]. This is partly due to the lack of technical facilities and awareness among the community. However, no such studies are available from Saudi Arabia, reporting these chemicals in indoor air and suspended particulate matter (PM). The monitoring of indoor environments for different pollutants is crucial for assessing exposure to the vulnerable exposure groups such as toddlers (hand-to-mouth behavior) and young children since they spend a lot of time indoors [13,16]. Analysis of indoors dust and atmospheric suspended fine particles is significant for this age group and especially in the Middle East region where due to outdoor weather conditions children stay indoor [17]. Indoor dust is considered an archive of indoor pollution that accumulates pollutants for a long time. Due to the hand-to-mouth and licking habits of young children involuntarily swallow and/or inhaling the varying amount of dust [11,17]. Due to the lack of moisture and sunlight in the indoor environment, many contaminants do not breakdown and show slow degradation [11]. Nevertheless, monitoring studies are very important for the detailed insights about the spatiotemporal occurrence trends of chemicals in the changing environment and to assess the effective implementation of new regulations to control their adverse impact on the environment and associated human population.

Saudi Arabia is going through fast industrialization for the past few decades Saudi population’s lifestyle. As a result, the lifestyle of the Saudi population has also changed dramatically. Still, studies are lacking to understand the impact of changing lifestyles and changing indoor environments on their health. Due to these scenarios, the current study reported these chemicals’ incidence into the suspended particulate material (PM10) of the Saudi indoor environment for the first time in Saudi Arabia. This primary study objective is to explore the profiling of selected OPEs and phthalates in PM10 and dust of selected Saudi indoors and estimate the exposed population’s exposure via dust ingestion, dermal intake and air inhalation.

## 2. Materials and Methods

### 2.1. Chemicals and Solvents

Analytical standards of phthalates namely bis(2-ethylhexyl) phthalate (DEHP), benzyl butyl phthalate (BzBP), bis(2-ethylhexyl) adipate (DEHA), dibutyl phthalate (DBP), diethyl phthalate (DEP), dimethyl phthalate (DMP) and OPEs namely tris (1,3-chloro-2-propyl) phosphate (TCPP), tris (2-chloroethyl) phosphate (TCEP), tris (1,3-dichloro-2-propyl) phosphate (TDCPP) and triphenyl phosphate (TPhP) were purchased from AccuStandards and Sigma Aldrich. Corresponding deuterated (d4-labelled) phthalates were used as internal standards (ISs) for phthalates while TCEP-d12 was used for OPEs. All stock solutions for the analytical standards were prepared in iso-octane and toluene. Acetone, dichloromethane (DCM), n-hexane (n-Hex) and iso-octane were of analytical grade obtained from Sigma Aldrich.

### 2.2. Sampling and Analysis 

For this study, paired particulate material (PM10) and indoor dust samples were collected simultaneously from households (*n* = 20), hotels (*n* = 10) and university offices (*n* = 10) of Jeddah, Saudi Arabia. The households sampled in this study were from Jeddah’s town, and they were all apartments that are the major part of housing in the city of Jeddah. This type of apartment/housing is typical and shared throughout the country. The sampled hotels were of 3- and 4-star categories that are usually used for most visiting pilgrims. Floor dust was obtained using the vacuum cleaner method previously reported by Ali et al. [18]. To have enough dust from hotel rooms, collected the sample after at least 20 h of the last room service. Dust samples were collected from the occupied room for overnight and before checking out. Briefly, the selected room from the participant household and hotel was vacuumed thoroughly for 5 min, and the accumulated dust was wrapped in aluminum foil and stored in a zipped bag. Before each sample the vacuum cleaner was thoroughly cleaned with solvent soaked tissue paper to avoid cross-contamination between the samples. For air-condition (AC) filter dust, AC filter was removed from the installed window and split AC, and dust attached with the filter was brushed off on aluminum foil, wrapped and kept in the zipped bag. Except for a couple of AC, most of the sampled AC were window AC. Thoroughly cleaned the brush used for the AC filter dust with solvent rinsed tissues before each sample to eliminate cross-contamination. The samples were transferred to the lab and were stored at −20 °C until analysis. To get homogenized dust samples for quantitative analysis, each dust sample was sieved using mesh (200 µm). An air sampler, Model 400 Micro-Environmental Monitor ^TM^, MSP Corporation, was used in the selected households (kitchen and living rooms), hotel rooms and university offices to collect PM10 samples. The sampler was installed for 24 h in each selected indoors, and the PM10 sample was collected on 47 mm quartz fiber filter paper using the flow of 10 L min^−1^. Before sampling filter papers were oven-baked at 400 °C for 6 h and kept in the desiccators until use. This precondition of the filter paper helped in getting rid of any moisture and prior contamination. A microbalance was used to measure PM10 levels, and after that, sampled filter paper stored at −20 °C until analysis.

### 2.3. Sample Preparation and Quantitative Analysis 

A detailed description of sample preparation is provided by Ali et al. [18]. Briefly, accurately measured dust (AC filter and floor dust) typically 50 mg was taken. After spiking with ISs, solvent mixture (hexane/acetone (4:1, *v*/*v*)) was added, and samples were extracted using ultrasonication (20 min) followed by centrifugation (3000 rpm for 10 min). The supernatant was collected in a clean tube, the same extraction procedure was repeated twice with the leftover samples. The extracts were pooled and brought to incipient dryness using a gentle stream of nitrogen. After drying, samples were resolubilized again in 1 mL the solvent mixture (hexane and toluene) and cleaned further using silica BondElut (500 mg, 3 mL) and 10 mL solvent mixture (hexane/dichloromethane) and after with 4 mL of ethyl acetate for quantitative analysis. After elution, the obtained fraction was concentrated to incipient dryness under a gentle stream of nitrogen. It then was resolubilized in 100 µL of iso-octane for gas chromatography–mass spectrometry (GC–MS) analysis. The same extraction procedure used for the extraction of OPEs and phthalates from PM10.

The detailed description of the used instrument analysis is provided elsewhere [17,19]. Briefly, for quantitative analysis, a TSQ™ 8000 Evo triple quadrupole GC-MS/MS (Thermo Fisher Scientific, Waltham, MA, USA) in the selected ion monitoring (SIM) mode used. A fused silica capillary column (TR5 30 M × 0.25 mm × 0.25 µm) used for the separation. The temperatures of the injector and ion source were 230 °C and 280 °C, respectively. Helium was used as the carrier gas at 1.5 mL min^−1^. The oven temperature was raised from 90 to 300 °C using a ramp of 15 °C min^−1^ and held for 1 min.

To have good QA/QC during the experimental procedure, all the glass-wares used were baked at 400 °C overnight and kept at 100 °C until use. Standard reference material (SRM) 2585 from the National Institute of Standards and Technology (NIST), procedural blanks (1 for every 8 samples), washed Na_2_SO_4_ (dust replica) spiked with a known concentration of standards were used to evaluate the procedure accuracy. The analytes’ levels found in procedural blanks corrected from the concentrations of the analysts in the samples. The values of OPEs and phthalates in SRM 2585 were similar (RSD < 25%) with other reported values [20] and other studies mentioned in Appendix A. Recovery of phthalates and OPEs in spiked Na_2_SO_4_ ranged between 70 and 130%. 

### 2.4. Human Risk Assessment Calculations 

In the present study, health risk assessment for the local population was calculated by per day exposure, incremental lifetime cancer risk (ILCR), hazard quotient (HQ) and hazardous index (HI). For this purpose, different equations were used as reported earlier [21,22]. Since no data on dust samples from the office was available and data from hotels was also limited, therefore for ILCR and HQ calculations, we only considered data from households on PM10 and dust.

The following Equations (1)–(3) were used to calculate non-carcinogenic chronic daily intake via dust ingestion, inhalation and dermal contact. For HQ calculation of each exposure route Equation (4) was used and HI was calculated by combining the HQ of different exposure routes (Equation (5)).
(1)$${\mathrm{CDI}}_{\mathrm{Ingestion-nca}}\;=\;C_{n\;}\times \frac{R_{\mathrm{ing}\;}\times \;\mathrm{EF}\;\times \;\mathrm{ED}}{\mathrm{BW}\;\times {\;\mathrm{AT}}_{\mathrm{nca}}}\times \;\mathrm{CF}$$
(2)$${\mathrm{CDI}}_{\mathrm{Inhalation-nca}}\;=\;C_{n}\times \frac{R_{\mathrm{inh}\;}\times \;\mathrm{EF}\;\times \;\mathrm{ET}\;\times \;\mathrm{ED}}{\mathrm{PEF}\;\times \;\mathrm{BW}\;\times {\;\mathrm{AT}}_{\mathrm{nca}}}$$
(3)$${\mathrm{CDI}}_{\mathrm{Dermal contact-nca}}\;=\;C_{n\;}\times \frac{\mathrm{SA}\;\times \;\mathrm{SL}\;\times \;\mathrm{ABSd}\;\times \;\mathrm{EF}\;\times \;\mathrm{ED}}{\mathrm{BW}\;\times {\;\mathrm{AT}}_{\mathrm{nca}}}\times \;\mathrm{CF}$$
HQ = CDI-nca/RfD (for each exposure route)(4)
HI = (HQ_ingestion_ + HQ_inhalation_ + HQ_dermal contact_)(5)

In the above equations, C_n_ represents the chemicals’ concentration, and for these calculations, we used the 90th percentile of the concentrations. Concentrations in dust were considered for ingestion and dermal contact while levels of these chemicals in PM10 were considered for the inhalation route. Since the AC filter dust is trapped in the filter and humans are not directly exposed to this dust. However, humans are directly and indirectly getting exposed to floor dust. Though, a part of AC filter dust, which escaped the AC filter, also settled down on the floor dust. Therefore, for these calculations, we only considered floor dust values relevant to ingestion and dermal exposure ro utes. The parameters of the Equations (1)–(3) are explained in Table 1. We considered high dust intake, for both adults and children, because of the dry and dusty environment of Saudi Arabia. While indoor Saudi public use air conditioning for cooling throughout the year, this results in more regular indoor air circulation and a high amount of fine dust particles [21]. 

We estimated carcinogenic risk exposure via different exposure routes using Equations (6)–(8) and the total carcinogenic risk was calculated combining all calculated exposure routes and cancer slope (SF) for children and for adults using the below Equation (9) [22].
(6)$${\mathrm{CDI}}_{\mathrm{Ingestion-ca}}\;=\;C_{n\;}\times \frac{{\mathrm{IR}}_{\;}\times \;\mathrm{EF}}{\mathrm{ATca}}\times \;\mathrm{CF}$$
(7)$${\mathrm{CDI}}_{\mathrm{Inhalation-ca}}\;=\;C_{n\;}\times \frac{\mathrm{EF}\;\times \;\mathrm{ET}\;\times \;\mathrm{ED}}{\mathrm{PEF}\;\times \;24\;\times {\;\mathrm{AT}}_{\mathrm{ca}}}\;\text{×}\;10^{3}$$
(8)$${\mathrm{CDI}}_{\mathrm{Dermal contact-ca}}\;=\;C_{n\;}\times \frac{\mathrm{ABSd}\;\times \;\mathrm{EF}\;\times {\;\mathrm{DFS}}_{\mathrm{adj}}}{{\mathrm{AT}}_{\mathrm{ca}}}\times \;\mathrm{CF}$$
(9)$$\mathrm{ILRC}\;=\;({\mathrm{CDI}}_{\mathrm{ingestion-ca}}\;\times \;\mathrm{SF}\;\mathrm{oral})\;+\;({\mathrm{CDI}}_{\mathrm{inhalation-ca}}\;\times \;\mathrm{SF}\;\mathrm{inhalation})\;+\;({\mathrm{CDI}}_{\mathrm{dermal contact-ca}}\;\times \;\mathrm{SF}\;\mathrm{dermal})$$

Cancer slope factor (SF) (mg kg^−1^ d^−1^) was available for only TCEP, DEHA and DEHP for oral and inhalation routes [17,19]. Where SF was not available for a specific route, we used available SF for all routes. 

Estimated daily intake (EDI) was calculated via dust ingestion and air inhalation using the following Equation (10):Estimated daily intake (ng per kg BW per day) = (C_n_ × I_R_/BW) × F_time_(10)
where C_n_ indicates the concentrations of chemicals in the dust (ng g^−1^) and PM10 (ng m^−3^), I_R_ is the dust ingestion rate (100 mg d^−1^ for adults and 200 mg d^−1^ for young children) and inhalation rate (20 m^3^ for adults and 7.6 m^3^ for young children, and F_time_ is the fraction of time people spend in households (24 h for people not working and 16 h (66%) for working people) and for people working in hotels (8 h, 33%). With the lack of knowledge on these chemicals’ bioaccessibility, we assumed 100% bioaccessibility for the EDI. Bodyweight of 70 kg for adults and 15 kg for young children was considered for the calculations.

### 2.5. Statistical Analysis

For descriptive analysis, Microsoft Excel 2007 was used. A two-sample *t*-test was applied using GraphPad to study the difference of PM10 in different microenvironments, and the significance level was *p* <0.05. Since the data did not show normal distribution, therefore data was log-transformed, and outliers were removed before using two-sample *t*-tests.

## 3. Results and Discussion

### 3.1. OPEs in Indoor PM10 and Dust

The total concentrations of OPEs (∑OPEs) in PM10 ranged between 9 and 295 ng m^−3^ (Table 2). Among various types of indoor microenvironments, PM10 samples collected from kitchen showed the highest OPEs concentrations (median: 70 ng m^−3^), followed by those from household living rooms (median: 68 ng m^−3^), hotel rooms (median: 62 ng m^−3^) and offices (median: 20 ng m^−3^). Levels of ∑OPEs were significantly (*p* < 0.05) lower in the PM10 collected from university offices than those of other studied indoor microenvironments. To the best of our knowledge, this is the first study reporting OPEs in PM10 from various microenvironments of this region. Among studied OPEs, TCPP was present at the highest concentrations in PM10 of all microenvironments, while TDCPP contributed the lowest proportions in the PM10 samples. Although OPEs in PM10 collected from the kitchen and living rooms of the households varied but were not significantly different (*p >* 0.05). Similarly, the profile of these chemicals was also similar in both microenvironments (Figure 1). This might indicate that the main sources of OPEs contamination inside the household rooms are the same, and cross ventilation between the rooms might be another plausible reason for similar chemicals. Among microenvironments, levels of TPhP were significantly higher (*p* < 0.05) in PM 10 samples from household than those found in the hotel and office, while levels of TCPP were significantly (*p* < 0.05) lower in the office PM10 than other indoor microenvironments. These significant differences indicate that the household air was more contaminated with OPEs than hotel and university offices. This showed there were more sources of emission inside the households than other studied microenvironments. Contrary to homes, most of the sampled hotel rooms and all the university offices had a centralized cooling and ventilation system, which might be another reason for the low concentrations of OPEs in these indoor air. The levels of OPEs in present indoor air samples were similar to more deficient than those reported in indoor air from other countries [12,15,27,28,29,30] (Appendix A). 

The total concentration of ∑OPEs in indoor dust samples of the AC filter, household floor and hotel rooms are presented in Table 3. TCEP (median: 14,480 ng g^−1^) was the major OPE in AC filter dust, while TCPP (median: 2650 ng g^−1^) and TDCPP (median: 1530 ng g^−1^) were the major OPEs in the hotel and household floor dust, respectively. The high levels of TDCPP in the households were due to the high concentration of this chemical in a couple of floor dust samples. By removing outliers, TCPP (median: 1300 ng g^−1^) was the major OPE in household floor dust as well. The levels of TCEP were significantly higher in AC filter dust than household and hotel floor dust, which might indicate material used in the air conditioning as a source of their emission. TDCPP was also significantly high (*p* < 0.05) in AC filter dust than household and hotel floor dust. Except for TCEP, the other three OPEs were detected in all the dust samples from studied microenvironments. The levels of OPEs in indoor dust were different from those reported from other countries (Appendix A), indicating different preferences to use chemicals used as FRs and plasticizers in various jurisdictions. 

The consistent occurrence trends for the high levels of OPEs in studied samples might indicate their widespread use as FRs and plasticizers in different consumer products. These chemicals reported to be used in flexible and rigid polyurethane foam, furniture foam, acrylic latexes for the back coating and binding of non-woven textiles products, leather, electronics and building construction laminates, furniture and baby products such as nursing pillows, portable mattresses, car seats, seat positioners and changing table pads [31,32]. Halogenated-OPEs have also been reported to exhibit high stability and are known to be relatively less degradable; this may show their gradual accumulation in the indoor environment [33]. Among halogenated-OPEs, TCEP was less frequently present in floor dust than other OPEs. The analyzed halogenated-OPEs have similar applications in consumer products. This may indicate that at TCPP and TDCPP might be preferred as a flame retardant in this region’s consumer products. TPhP has a wide range of applications and utilized for both purposes, i.e., as a flame retardant (Firemaster^®^ 550 (Chemtura, Philadelphia, PA, USA), electronic types of equipment, PVC, glues, nail polishes, etc.) and a plasticizer in many products. After the phasing-out of Penta-BDE, the Firemaster^®^ 550 (Chemtura) was used in polyurethane foam and other applications, resulting in its widespread occurrence in indoor environments found in this study too.

### 3.2. Phthalates in Indoor PM10 and Dust

The total concentrations of (∑phthalates) in PM10 ranged between 245 and 1900 ng m^−3^ (Table 2). To the best of our knowledge, this is the first study reporting phthalates in indoor air from various microenvironments of this region. Among different types of indoor microenvironments, PM10 samples collected from a kitchen contained the highest phthalates concentrations (median: 1000 ng m^−3^), followed by those collected from household living rooms (median: 830 ng m^−3^), hotel rooms (median: 650 ng m^−3^) and offices (median: 315 ng m^−3^). Levels of ∑phthalates were significantly lower in office PM10 (*p* < 0.05) than other studied indoor microenvironments. As discussed in the previous section (Section 3.1) centralized cooling and ventilation system, the low concentrations of phthalates in the air of these university offices and hotel rooms might be the reason. DEHP was present among studied phthalates at the highest concentrations in PM10 of all microenvironments, followed by DBP > DZBP > DEP > DEHA > DMP. DMP contributed the least proportions in the collected PM10 samples and was detected in <10% of PM10 samples of each indoor environment (Figure 1), which is due to the volatile nature of the DMP. Levels of phthalates measured into PM10 of the household’s kitchen and living rooms varied; however, there was no statistically significant difference (*p* > 0.05). DEP levels were significantly lower in homes (living room and kitchen) PM10 than those found in hotels and offices. DEP was used to improve the performance and durability of several products. As a plasticizer, it was added to plastic polymers to help maintain flexibility. It has been used in various products, including plastic films, rubber, tape, toothbrushes, automotive components, tool handles and toys. In addition to plastics, DEP is present in a wide range of personal care products (e.g., cosmetics, perfume, hair spray, nail polish, soap, detergent and lotions), industrial materials (e.g., rocket propellant, dyes, packaging, sealants and lubricants) and medical products (e.g., enteric coatings on tablets and in dental impression materials) [5,34]. These everyday products treated with DEP are commonly used in households as well. Therefore, no supporting information could explain why these levels were high in the office and hotel PM10 than homes. More detailed studies with more details are needed to understand their higher occupational settings occurrence than residential settings. DEHP levels were significantly lower (*p* > 0.05) in PM10 samples from offices than other sampled microenvironments. Although less volatile nature of DEHP, the high levels of its presence in PM10 than some of the other analyzed phthalates can be justified due to their increased fact in indoor dust, indicating extensive use in indoor consumer products. Number of studies have reported the concentrations of phthalates in the indoor environment [12,14], and those reported concentrations were similar and/or lower than those found in this study (Appendix A). 

The total concentrations of ∑phthalates in indoor dust samples from different microenvironments are presented in Table 3. DEHP was the primary phthalate in all three samples microenvironments, i.e., AC filter (median: 671,750 ng g^−1^), hotel floor (median: 745,500 ng g^−1^) and household floor dust (median: 573,100 ng g^−1^). DBP was the 2nd significant phthalate in all indoor microenvironment dust with a median concentration of 18, 850 ng g^−1^ in household floor dust, 41,450 ng g^−1^ in hotel floor dust and 26,700 ng g^−1^ in AC filter dust. The high DEHP levels were not surprising considering its extensive application as a plasticizer in polymer products, which are extensively utilized in furniture material, cosmetics and personal care products [35]. DBP is a vital plasticizer used in extensive engineering plastics, e.g., PVC, widely used in plumbing and other corrosive materials. Chemicals are added in these products and gradually released from them into the surrounding environments, and it may result in extensive contamination of indoor dust. The DEHA was the 3^rd^ dominant phthalate in dust samples from all three microenvironments with median concentrations of 3550, 7810 and 1620 ng g^−1^ of dust in AC filter, hotel and household floor dust, respectively. DEHA were significantly higher (*p* < 0.05) in hotel floor dust than the AC filter dust, indicating AC is not a primary source of DEHA in indoors. Still, other indoor inventories are mainly responsible for their indoor presence. The major application of DEHA is in hydraulic fluid and PVC-based plastic food wrap but is not chemically bound to the polymer, thus migrates into the surrounding environment [35]. The higher levels of DEHA in kitchen PM10 than other indoors were probably associated with food wraps. However, indoor dust from the different indoor environment also indicates its applications in other consumer products responsible for their indoor environment presence. DEP was also detected in dust from all microenvironments with median values >1000 ng g^−1^. DMP contributed the least with the highest median concentration of 410 ng g^−1^ of dust in the household floor dust, and this is understandable because of the volatile nature of DMP.

Phthalates contamination has been reported into indoor dust from different countries (Appendix A). The variation in the profile and indoor occurrence of phthalates may indicate the regulations of their use in various jurisdictions. DEHP was the dominant phthalates in all studies except Bulgarian house dust, where DBP concentration was highest among studied phthalates (Appendix A). DEHP contributed > 90% in phthalate profile of dust samples (Figure 2) collected from all microenvironments in the current study, which is in agreement to those reported into indoor dust from Denmark, Finland, France, Spain, UK, China, USA, Germany, Japan, Kuwait, Norway and Taiwan (Appendix A).

### 3.3. Human Risk Assessment

Several studies have reported that exposure to OPEs and phthalates is associated with various health issues such as DEHP, an endocrine disruptor and have carcinogenic properties [36]. Rowdhwal and Chen [35] have reviewed the toxicities of DEHP and discussed the health implication due to its exposure, e.g., testicular, ovarian, endometrial, neuro-, cardio- and hepatotoxicity. Other phthalates also reported causing potential health effects such as low BMI, increased organ weight, endocrine disruptor, teratogenic, developmental and reproductive effects [34]. Like phthalates, exposure to OPEs is also linked to many health problems. Several studies are available in the literature, showing that OPEs are endocrine disruptive and exhibit carcinogenic properties [37,38,39]. To investigate the possible health risk associated with the long term and daily exposure of studied chemicals different exposure parameters were calculated for various exposure routes and different Equations (1)–(10) as reported above in the methodology chapter. EDI was multifold below reference dose (RfD) values for all selected chemicals for exposure from dust and PM10 (Table 4). However, estimated EDI for DEHP was half the value of RfD for young children (Table 4). This showed indoor exposure to DEHP via dust ingestion, and PM10 inhalation is of concern for the young children, especially with their developing body and immune system. To further study the non-carcinogenic risk to the Saudi young children and adults, the HQ and HI was calculated using Equations (1)–(5). The estimated HQ and HI of individual chemicals except for DEHP from dust ingestion, inhalation, and dermal contact were well within the acceptable level (<1). However, HQ via dust ingestion and HI was >2 for DEHP for young children exposure (Table 5). This showed that non-carcinogenic risk is low from most of the analyzed chemicals except DEHP.

As discussed above, many of the studied chemicals reported to be carcinogenic; therefore, the main interest in calculating the ILCR was to look at the potential long-term cancer risk via dust exposure to the Saudi young and adult people from the indoors. The probabilistic ILCR assessment was highest via dust ingestion, followed by dermal contact and inhalation (Table 5). The USEPA recommended a safe limit for long term cancer risk between 1.00 × 10^−4^ and 1.00 × 10^−6^ [40]. Therefore, for all the studied chemicals, estimated ILCR was within the USEPA recommendation range, which indicates a limited to moderate risk to the Saudi population from these chemicals to develop cancer via dust exposure from their indoors. However, ILCR calculated for DEHP and ∑phthalates via indoor dust exposure was 1.5 × 10^−5^, this indicates a moderate carcinogenic risk for the public health according to Health Canada Guidelines [41]. In a recent study, high levels of DEHP metabolites were reported in urine samples from young Saudi children from Riyadh, which found a positive correlation with oxidative stress and exposure to indoor dust was suggested as one of the significant exposure pathways [42]. Limited to no information available on the production and use of DEHP in Saudi Arabia, which is a cause of concern. According to EU REACH legislation DEHP is a Category 1B reprotoxic [43] and its use is highly regulated. Our calculated assessment showed that people living in urban Jeddah are exposed to phthalates and OPEs via dust and PM10 with a potential risk to their health. However, it is stressed to be cautioned to draw any definite conclusion because of the small samples analyzed. Although this study had limitations due to the limited data set, it indicates this study has limitations due to the limited data set. It shows the possible range of exposure to these chemicals from their indoors and the Saudi population. It highlights the importance of indoor monitoring for organic pollutants.

## 4. Conclusions

This is the first study reporting OPEs and phthalates in the indoor PM10 of Saudi Arabia. The ∑phthalates and ∑OPEs concentrations in PM10 from the kitchen were highest, which highlighted the risk for people working in the kitchen for a longer time. Dust from the AC filter was more contaminated with these chemicals than dust from other microenvironments. The air conditioning system (AC window and split AC) helps in indoor air circulation, but at the same time, it might circulate more chemicals in the indoor air, which otherwise settle down to the ground. However, large scale studies are needed to establish the overall contribution of different air conditioning systems to indoor pollution. DEHP was the major pollutant among studied chemicals in both PM10 and indoor dust from all microenvironments, which indicates its high use in consumer products in the local market. The EDI and ILCR calculations showed a cause of concern to the Saudi public health from exposure to DEHP via contaminated indoor dust and PM10. Dust ingestion and dermal contact were the primary exposure routes for both adults and young children. The probabilistic cancer risk assessment from ∑phthalates and DEHP, total ILCR, was high up to 1.5E-5 for Saudi adults, which might indicate a moderate risk to the exposed Saudi population and a cause of concern public health. Limited studies are available on this topic from this region. Therefore, extensive scale studies will be needed to understand the dynamics of organic pollution in Saudi indoor environments and their impact on public health.

## Figures and Tables

**Figure 1 ijerph-18-02125-f001:**
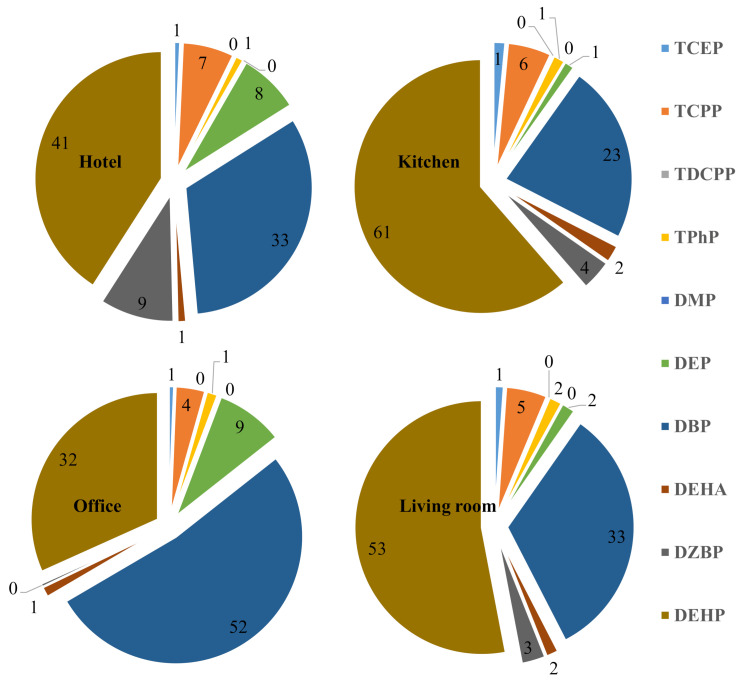
Contribution (%) of analyzed chemicals in PM10 of sampled indoor microenvironments of Saudi Arabia.

**Figure 2 ijerph-18-02125-f002:**
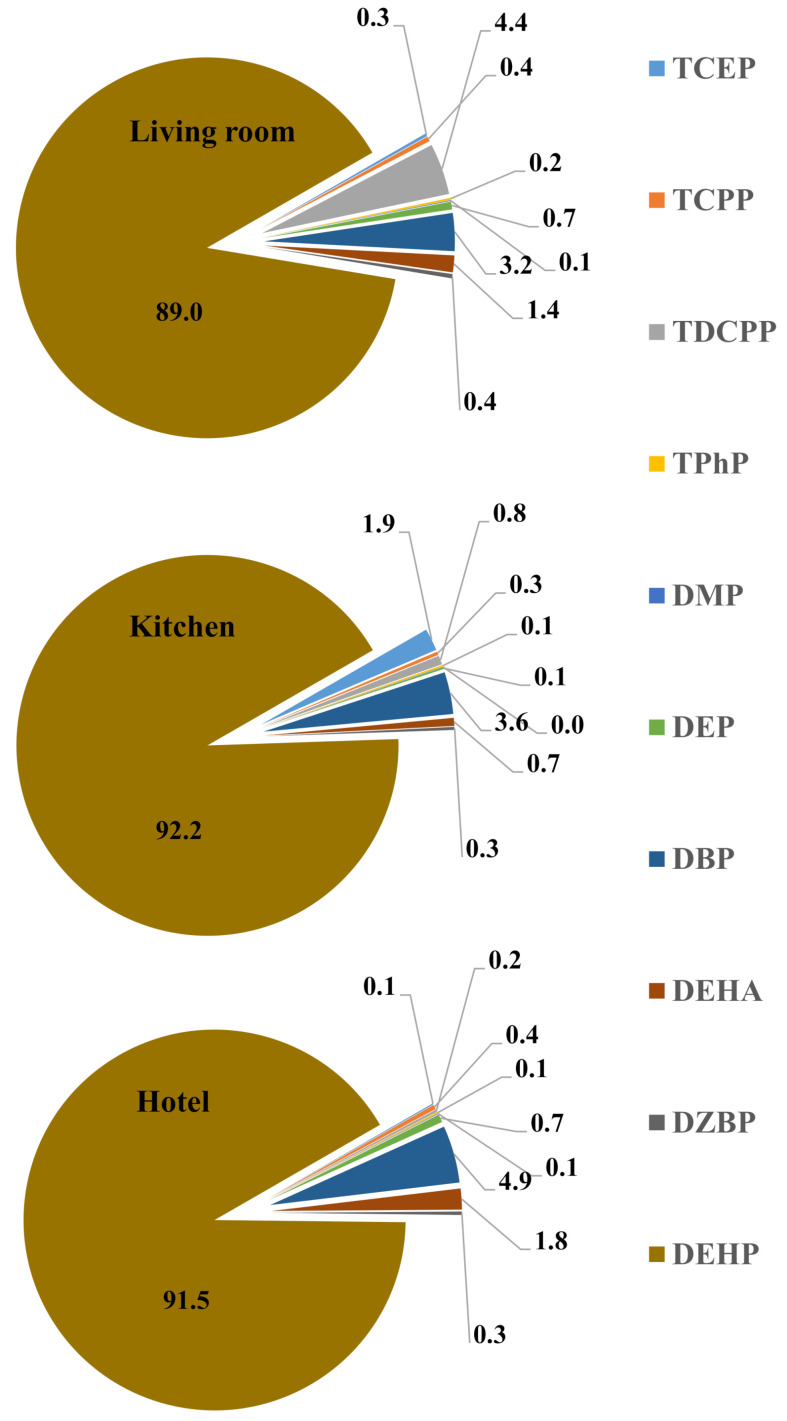
Contribution (%) of analyzed chemicals in indoor dust of sampled indoor microenvironments of Saudi Arabia.

**Table 1 ijerph-18-02125-t001:** Parameters used in human risk assessment equations.

Parameters	Children	Adults	Reference
Ingestion rate (R_ing_) (mg d^−1^)	200	100	[18]
Inhalation rate (R_inh_) (m^3^ d^−1^)	7.6	20	[18]
Exposure frequency (EF) (d year^−1^)	350		[23]
Duration of exposure (ED) (years)	2	30	[24]
Exposed skin area (SA) (cm^3^)	1600	6700	[24]
Dust to skin adherence factor (SL) (mg cm^−2^)	0.5		[24]
Dermal absorption factor (ABS_d_)	0.03	0.001	[23]
Particle emission factor (PEF) (m^3^ kg^−1^)	1.36 × 10^9^		[23]
Body weight (BW) (kg)	15	70	[25]
Lifetime (LT) (years)	70		[26]
Conversion factor (CF)	1 × 10^−6^		[23]
Dust dermal contact factor -age-adjusted (DFS_adj_) (mg × year kg^−1^ d^−1^)	362.4		[23]
Dust ingestion rate age-adjusted (IR) (mg × year kg^−1^ d^−1^)	113		[23]
Exposure time (ET) (h d^−1^)	17.8	20	[24]
Average non-carcinogenic exposure time (AT_nca_)	ED × 365		[23]
Average carcinogenic exposure time (AT_ca_)	LT × 365		[23]

**Table 2 ijerph-18-02125-t002:** Concentrations (ng m^−3^) of analyzed OPEs and phthalates in PM10 collected from different indoor microenvironments of Saudi Arabia.

Analytes	Kitchen (*n* = 15)		Hotel (*n* = 10)		Living room (*n* = 20)		Office (*n* = 10)	
	Mean ± StDev	Median (Min–Max)	Mean ± StDev	Median (Min–Max)	Mean ± StDev	Median (Min–Max)	Mean	Median (Min–Max)
TCEP	17 ± 25	6 (<LOQ–2)	6 ± 6	4 (LOQ–17)	11 ± 14	7 (<LOQ–54)	2 ± 4	<LOQ (<LOQ–10)
TCPP	63 ± 62	43 (7–180)	55 ± 58	42 (7–185)	51 ± 45	38 (3–155)	13 ± 8	12 (3–25)
TDCPP	2 ± 4	0.1 (<LOQ–17)	<LOQ	<LOQ (<LOQ–2)	1 ± 1	<LOQ (<LOQ–5)	<LOQ	<LOQ
TPhP	17 ± 12	15 (<LOQ–38)	8 ± 8	7 (<LOQ–25)	16 ± 30	9 (<LOQ–138)	5 ± 4	3 (2–12)
∑OPEs	97 ± 70	70 (9–225)	70 ± 62	62 (10–210)	78 ± 65	68 (8–295)	20 ± 8	20 (14–30)
DMP	<LOQ	<LOQ	1 ± 1	<LOQ (<LOQ–2)	<LOQ	<LOQ	<LOQ	<LOQ
DEP	14 ± 13	9 (3–48)	65 ± 52	62 (6–150)	17 ± 14	11 (3 ± 55)	30 ± 8	28 (25–45)
DBP	257 ± 280	191 (11–1130)	275 ± 200	255 (1–645)	320 ± 270	250 (17–1010)	180 ± 60	175 (100–260)
DEHA	25 ± 30	14 (4–90)	9 ± 7	6 (2–22)	15 ± 14	10 (2 ± 57)	5 ± 2	4 (3–7)
DZBP	44 ± 160	1 (<LOQ–600)	80 ± 220	1 (1–625)	30 ± 125	1 (<LOQ–585)	1 ± 1	2 (1–2)
DEHP	695 ± 340	610 (100–1150)	345 ± 300	230 (145–1060)	520 ± 260	520 (160–950)	111 ± 15	110 (90–125)
∑Phthalates	1030 ± 395	1000 (275–1700)	710 ± 315	650 (325–1180)	895 ± 360	830 (395–1900)	330 ± 70	315 (245–420)

**Table 3 ijerph-18-02125-t003:** Concentrations (ng g^−1^) of analyzed OPEs and phthalates in dust collected from different indoor microenvironments of Saudi Arabia.

Analytes	AC Filter Dust (*n* = 20)		Hotel Floor Dust (*n* = 10)		Household Floor Dust (*n* = 20)	
	Mean ± StDev	Median (Min–Max)	Mean ± StDev	Median (Min–Max)	Mean ± StDev	Median (Min–Max)
TCEP	16,100 ± 16,600	14,480 (<LOQ–58,200)	800 ± 1190	<LOQ (<LOQ–3370)	2500 ± 6500	280 (<LOQ–25,500)
TCPP	2500 ± 1800	1750 (670–5850)	2900 ± 2400	2650 (150–7800)	4100 ± 5400	1300 (25–17,300)
TDCPP	6500 ± 5850	5950 (650–19,700)	1600 ± 1800	660 (400–5650)	42,700 ± 157,800	1530 (<LOQ–613,000)
TPhP	1190 ± 550	920 (790–2500)	900 ± 250	835 (670–1540)	1730 ± 2490	800 (610–9000)
∑OPEs	263,00 ± 18,100	26,250 (3100–69,200)	6230 ± 4350	4632 (1860–16,000)	53,700 ± 162,200	4800 (650–615300)
DMP	360 ± 70	340 (280–500)	400 ± 85	410 (315–550)	710 ± 1100	290 (200–3950)
DEP	1250 ± 770	1140 (390–2950)	5090 ± 3030	4150 (1380–9000)	6455 ± 14,520	1020 (<LOQ–55,800)
DBP	30,400 ± 15,600	26,700 (13,900–56,400)	36,450 ± 18,650	41,450 (4350–57,650)	104,500 ± 242,000	18,850 (<LOQ–952,300)
DEHA	5850 ± 6300	3550 (1850–22,550)	13,350 ± 11,900	7810 (400–34,900)	14,000 ± 45,800	1620 (<LOQ–179,400)
DZBP	2250 ± 3750	930 (210–12,650)	2200 ± 2450	750 (230–5550)	3650 ± 10,550	460 (90–41,400)
DEHP	784,100 ± 459,500	671750 (391,900–1,844,600)	684,400 ± 396,900	745,500 (61,050–1,335,500)	871,700 ± 1,022,900	573,100 (350–3,550,300)
∑Phthalates	824,100 ± 467,500	720,550 (423,100–1,904,500)	741,900 ± 427,500	776,630 (70,400–1,441,700)	784,100 ± 986,800	475,900 (650–3,901,500)

**Table 4 ijerph-18-02125-t004:** Estimated daily exposure (ng kg^−1^ bw d^−1^) to OPEs and phthalates via dust ingestion and PM10 inhalation for Saudi young children and adults from their households.

Analytes	RfD	PM10			Floor Dust			Inhalation + Ingestion		
		Adult Hotel	Adult Households	Young Children Households	Adult Hotel	Adult Households	Young Children Households	Adult Hotel	Adult Households	Young Children Households
TCEP	22,000	2	3	6	3	4	30	5	6	35
TCPP	80,000	13	12	30	5	6	49	18	18	80
TDCPP	15,000	0	0	1	40	60	515	42	60	520
TPhP	70,000	3	4	9	2	2	20	5	6	30
DMP	100,000	0	0	1	1	1	9	1	1	9
DEP	800,000	8	4	10	9	9	80	16	15	90
DBP	100,000	75	75	180	115	150	1265	190	225	1450
DEHA	600,000	3	4	8	20	20	170	23	25	180
DZBP	200,000	11	7	15	4	5	45	16	10	60
DEHP	20,000	110	125	290	1150	1245	10,550	1260	1370	10,850

Reference dose (RfD) for phthalates [26] and OPEs [16].

**Table 5 ijerph-18-02125-t005:** Incremental lifetime cancer risk (ILCR) calculated using 90th percentile values of OPEs and phthalates in indoor dust for Saudi young children and adults from their households.

	Adults			Children				
**Non-Carcinogenic**	**CDInca-Ingestion**	**CDInca-Dermal**	**CDInca-Inhalation**	**CDInca-Ingestion**	**CDInca-Dermal**	**CDInca-Inhalation**		
TCEP	9.1 × 10^−6^	6.1 × 10^−7^	1.4 × 10^−10^	1.7 × 10^−4^	2.0 × 10^−5^	2.2 × 10^−10^		
TCPP	1.1 × 10^−5^	7.6 × 10^−7^	5.3 × 10^−10^	2.1 × 10^−4^	2.5 × 10^−5^	8.4 × 10^−10^		
TDCPP	1.7 × 10^−4^	1.2 × 10^−5^	1.6 × 10^−11^	3.2 × 10^−3^	3.8 × 10^−4^	2.5 × 10^−11^		
TPhP	5.1 × 10^−6^	3.4 × 10^−7^	1.1 × 10^−10^	9.6 × 10^−5^	1.2 × 10^−5^	1.7 × 10^−10^		
DMP	2.2 × 10^−6^	1.5 × 10^−7^	0.0 × 10^0^	4.1 × 10^−5^	4.9 × 10^−6^	0.0 × 10^0^		
DEP	2.3 × 10^−5^	1.5 × 10^−6^	1.7 × 10^−10^	4.3 × 10^−4^	5.1 × 10^−5^	2.7 × 10^−10^		
DBP	4.8 × 10^−5^	3.2 × 10^−6^	3.7 × 10^−9^	9.0 × 10^−4^	1.1 × 10^−4^	5.8 × 10^−9^		
DEHA	5.3 × 10^−5^	3.6 × 10^−6^	1.6 × 10^−10^	9.9 × 10^−4^	1.2 × 10^−4^	2.5 × 10^−10^		
DZBP	1.4 × 10^−5^	9.4 × 10^−7^	1.6 × 10^−11^	2.6 × 10^−4^	3.1 × 10^−5^	2.5 × 10^−11^		
DEHP	2.1 × 10^−3^	1.4 × 10^−4^	3.6 × 10^−9^	4.0 × 10^−2^	4.8 × 10^−3^	5.7 × 10^−9^		
**Hazardous Index**	**HQ-ingestion**	**HQ-dermal**	**HQ-inhalation**	**HQ-ingestion**	**HQ-dermal**	**HQ-inhalation**	**HI (Adult)**	**HI (Children)**
TCEP	4.1 × 10^−4^	2.8 × 10^−5^	6.4 × 10^−9^	7.7 × 10^−3^	9.3 × 10^−4^	1.0 × 10^−8^	4.4 × 10^−4^	8.7 × 10^−3^
TCPP	1.4 × 10^−4^	9.5 × 10^−6^	6.6 × 10^−9^	2.7 × 10^−3^	3.2 × 10^−4^	1.0 × 10^−8^	1.5 × 10^−4^	3.0 × 10^−3^
TDCPP	1.1 × 10^−2^	7.7 × 10^−4^	1.1 × 10^−9^	2.1 × 10^−1^	2.6 × 10^−2^	1.7 × 10^−9^	1.2 × 10^−2^	2.4 × 10^−1^
TPhP	7.3 × 10^−5^	4.9 × 10^−6^	1.6 × 10^−9^	1.4 × 10^−3^	1.6 × 10^−4^	2.5 × 10^−9^	7.8 × 10^−5^	1.5 × 10^−3^
DMP	2.2 × 10^−5^	1.5 × 10^−6^	0.0 × 10^0^	4.1 × 10^−4^	4.9 × 10^−5^	0.0 × 10^0^	2.3 × 10^−5^	4.6 × 10^−4^
DEP	2.9 × 10^−5^	1.9 × 10^−6^	2.1 × 10^−10^	5.3 × 10^−4^	6.4 × 10^−5^	3.3 × 10^−10^	3.0 × 10^−5^	6.0 × 10^−4^
DBP	4.8 × 10^−4^	3.2 × 10^−5^	3.7 × 10^−8^	9.0 × 10^−3^	1.1 × 10^−3^	5.8 × 10^−8^	5.1 × 10^−4^	1.0 × 10^−2^
DEHA	8.9 × 10^−5^	5.9 × 10^−6^	2.7 × 10^−10^	1.7 × 10^−3^	2.0 × 10^−4^	4.2 × 10^−10^	9.5 × 10^−5^	1.9 × 10^−3^
DZBP	7.0 × 10^−5^	4.7 × 10^−6^	8.1 × 10^−11^	1.3 × 10^−3^	1.6 × 10^−4^	1.3 × 10^−10^	7.5 × 10^−5^	1.5 × 10^−3^
DEHP	1.1 × 10^−1^	7.2 × 10^−3^	1.8 × 10^−7^	2.0 × 10^0^	2.4 × 10^−1^	2.9 × 10^−7^	1.1 × 10^−1^	**2.2 × 10^0^**
**Carcinogenic**	**CDIca-Ingestion**	**CDIca-Dermal**	**CDIca-Inhalation**	**CDIca-Ingestion**	**CDIca-Dermal**	**CDIca-Inhalation**	**ILRC (Adult)**	**ILRC (Children)**
TCEP	2.1 × 10^−5^	6.6 × 10^−8^	8.8 × 10^−9^	2.1 × 10^−5^	2.0 × 10^−6^	5.2 × 10^−10^	4.1 × 10^−7^	4.8 × 10^−8^
DEHA	1.2 × 10^−4^	3.9 × 10^−7^	1.0 × 10^−8^	1.2 × 10^−4^	1.2 × 10^−5^	6.0 × 10^−10^	1.4 × 10^−7^	1.4 × 10^−8^
DEHP	4.8 × 10^−3^	1.5 × 10^−5^	2.3 × 10^−7^	4.8 × 10^−3^	4.6 × 10^−4^	1.4 × 10^−8^	**1.5 × 10^−5^**	1.5 × 10^−6^

Note: Bold values indicate the cause of concern.

## Data Availability

All of the data is presented in the article and associated Appendix A.

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
