# Peer review of "Exposure to Phthalate and Organophosphate Esters via Indoor Dust and PM10 Is a Cause of Concern for the Exposed Saudi Population"

_ijerph, 2021, doi:10.3390/ijerph18042125_

Round 1

Reviewer 1 Report

This work investigated the occurrence of organophosphate esters and phthalates in the many kinds of dusts and the hazardous risk to a person, Saudi Arabia. The results are very interesting and helpful for the hazardous risk assessment of organophosphate esters and phthalates. However, I recommend that authors should revise some points. The manuscript can be accepted after revision. The modified points are shown as follows.

1) line.1, p4

This line should be left-justified.

2) 2.1. Chemicals and solvents, p2

“Tris (1,3-dichloro-2propyl) phosphate” should be changed to “Tris (1,3-dichloro-2-propyl) phosphate”.

3) Table1

The parameters should be left-justified.

“Children” should be made one line.

Table 1 should be fit on one page.

4) Table2

“ΣOPEs” should be centered.

“ΣPhthalates” should be made one line.

“Median” under Office (n=10) should be changed to “Median (Mini-Max)”.

5) Table 4

The bold “TCEP” should be changed to the normal “TCEP”.

The note should be left-justified.

Author Response

We thank the reviewers for their thorough reading of the manuscript and their suggestions for improvement. We are pleased to see that the reviewers have discussed the critical points and suggested essential enhancements to improve the manuscript's quality. In line with the reviewer’s suggestions, we have revised the manuscript and marked with track changes in the text in the revised version of the manuscript. Therefore, we have considered all objections and suggestions (marked with track changes in the text in the revised version of the manuscript). Please find here below a point-by-point reply to the reviewers’ comments. (R – 'reviewers' comments; A – 'authors' comments, Changes performed in the manuscript).

This work investigated the occurrence of organophosphate esters and phthalates in the many kinds of dusts and the hazardous risk to a person, Saudi Arabia. The results are very interesting and helpful for the hazardous risk assessment of organophosphate esters and phthalates. However, I recommend that authors should revise some points. The manuscript can be accepted after revision. The modified points are shown as follows.

Response: Thank you very much for your observations. Necessary changes have been made in the text as per the reviewer comments.

1) Line.1, p4, This line should be left-justified.

Response: Agreed and modified in the revised manuscript. 

2) 2.1. Chemicals and solvents, p2

“Tris (1,3-dichloro-2propyl) phosphate” should be changed to “Tris (1,3-dichloro-2-propyl) phosphate”.

Response: Thanks for your remarks. Correction carried out in the revised manuscript.

3) Table1, The parameters should be left-justified. “Children” should be made one line. Table 1 should be fit on one page.

Response: Agreed and corrected in the revised manuscript.

4) Table2, “ΣOPEs” should be centred. “ΣPhthalates” should be made one line.

“Median” under Office (n=10) should be changed to “Median (Mini-Max)”.

Response: Thanks and changed as suggested.

5) Table 4, The bold “TCEP” should be changed to the normal “TCEP”. The note should be left-justified.

Response: Thanks, corrected as suggested.

Reviewer 2 Report

This paper reports on phthalate and organophosphate esters in different indoor environments in Saudi Arabia.The authors correctly state that few measurements of this type have been reported, so this paper is a good start to determining how these chemicals affect both children and adults. They also correctly state that this study is not extensive enough to provide firm conclusions.

I have some questions and suggestions about this work that should be considered.

  1. It is difficult to determine what types of homes were sampled. Are these homes similar to each other in terms of age, size, types of AC systems, number of inhabitants, and the socio-economic status? What fraction of homes in Saudi Arabia are similar? I realize that different countries (and journals) have different rules regarding on what information can be reported when dealing with human subjects, so it may not be possible to answer all of these questions. However, it would be good to know if these homes are typical or not. I am making the assumption that hotels and university offices are more uniform across the developed countries of the world, but I am not certain if that is true.
  2. In the sampling section, the type of AC system in the homes should be described.
  3. The statement about how hotel rooms were sampled was a bit confusing to me, especially the phrase "20 hours of room service". Does that mean the room was occupied for 20 hours, or was the room empty after the room was cleaned (or serviced as the term is used in the US).
  4. Brushing off the AC filters does not seem to be a quantitative method for collecting particles. Is there any data comparing brushing with a more extensive method of extraction of the materials of concern here?
  5. On page 7, two samples had high levels of TDCPP, and this data was treated as an outlier. What was the justification? Was there anything different about this house or its use that might explain the higher levels?

Comments:

  1. I find the Tables S2, S3, and S4 to be an interesting comparison of these results to other countries. Could they be be moved into the main paper?
  2. In discussing different levels of different compounds in the three different environments, there is significant speculation about the sources of these chemicals. Some seems rather unlikely (e.g., rocket propellants, hydraulic fluids), and others have a wide range of more commonly used products. Since this paper is not attempting to determine sources, this discussion can be shortened.

Author Response

We thank the reviewers for their thorough reading of the manuscript and their suggestions for improvement. We are pleased to see that the reviewers have discussed the critical points and suggested essential enhancements to improve the manuscript's quality. In line with the reviewer’s suggestions, we have revised the manuscript and marked with track changes in the text in the revised version of the manuscript. Therefore, we have considered all objections and suggestions (marked with track changes in the text in the revised version of the manuscript). Please find here below a point-by-point reply to the reviewers’ comments. (R – 'reviewers' comments; A – 'authors' comments, Changes performed in the manuscript).

Reviewer 2:

This paper reports on phthalate and organophosphate esters in different indoor environments in Saudi Arabia. The authors correctly state that few measurements of this type have been reported, so this paper is a good start to determining how these chemicals affect both children and adults. They also correctly state that this study is not extensive enough to provide firm conclusions.

I have some questions and suggestions about this work that should be considered.

  1. It is difficult to determine what types of homes were sampled. Are these homes similar to each other in terms of age, size, and types of AC systems, number of inhabitants, and the socio-economic status? What fraction of homes in Saudi Arabia are similar? I realize that different countries (and journals) have different rules regarding on what information can be reported when dealing with human subjects, so it may not be possible to answer all of these questions. However, it would be good to know if these homes are typical or not. I am making the assumption that hotels and university offices are more uniform across the developed countries of the world, but I am not certain if that is true.

Response: Yes, university offices and hotels across the developed countries are the same with some adjustments according to the local culture and environment. The households sampled in this study were from Jeddah's town, and they were all apartments that are the major part of housing in the city of Jeddah, we mentioned this in the revised manuscript. This type of apartment/ housing is typical and shared throughout the country. It was also much more convenient to do sampling in apartments than villas because villas are not standard and prominent families living in estates in many sampled districts. Hence, it is not easy to convince them to participate in the study. 

  1. In the sampling section, the type of AC system in the homes should be described.

Response: Provided in the revised manuscript, “For air-condition (AC) filter dust, AC filter was removed from the installed window and split AC, and dust attached with the filter was brushed off on aluminium foil, wrapped and kept in the zipped bag. Except for a couple of AC, most of the sampled AC were window AC. Thoroughly cleaned the brush used for the AC filter dust with solvent rinsed tissues before each sample to get rid of cross-contamination.”

  1. The statement about how hotel rooms were sampled was a bit confusing to me, especially the phrase "20 hours of room service". Does that mean the room was occupied for 20 hours, or was the room empty after the room was cleaned (or serviced as the term is used in the US)

Response: To have enough dust from hotel rooms collected the sample after at least 20 h of last room service. Yes, dust samples were collected from the occupied room for overnight and before checking out.

  1. Brushing off the AC filters does not seem to be a quantitative method for collecting particles. Is there any data comparing brushing with a more extensive method of extraction of the materials of concern here?

Response: We did not find the certified method for a dust sample collection from AC systems. Most of the dust that passes through the AC system is filtered and trapped in the AC filter. Brushing off the dust from filter gives us the maximum amount of dust from the AC filter, as we previously used in some other studies (https://doi.org/10.1016/j.scitotenv.2019.133995; https://doi.org/10.1016/j.scitotenv.2016.06.093).  

  1. On page 7, two samples had high levels of TDCPP, and this data was treated as an outlier. What was the justification? Was there anything different about this house or its use that might explain the higher levels?

Response: Grubb’s test removed outliers by using GraphPad to have normalized the data. The collected information during sampling as not enough to determine reasons for such high levels in those couple of samples; therefore, we did not speculate them in the manuscript.    

Comments:

  1. I find the Tables S2, S3, and S4 to be an interesting comparison of these results to other countries. Could they be moved into the main paper?

Response: Agreed that these tables have essential information, but this information was not the study's primary focus. Therefore, these tables are also suitable for supplementary information. We already have five tables and two figures in the main manuscript, which have more relevant details for the main manuscript.  

  1. In discussing different levels of different compounds in the three different environments, there is significant speculation about the sources of these chemicals. Some seems rather unlikely (e.g., rocket propellants, hydraulic fluids), and others have a wide range of more commonly used products. Since this paper is not attempting to determine sources, this discussion can be shortened.

Response: Agreed, and we removed some of the information to shorten the discussion.
